# GPT-4 as an Effective Zero-Shot Evaluator for Scientific Figure Captions

**Ting-Yao Hsu,[1] Chieh-Yang Huang,[1] Ryan Rossi,[2] Sungchul Kim,[2]**
**Clyde Lee Giles,[1] Ting-Hao 'Kenneth' Huang[1]**
[1]Pennsylvania State University, University Park, PA, USA.
`{txh357,chiehyang,clg20,txh710}@psu.edu`
[2]Adobe Research, San Francisco, CA, USA.
`{ryrossi,sukim}@adobe.com`

## Abstract

There is growing interest in systems that generate captions for scientific figures. However, assessing these systems' output poses a significant challenge. Human evaluation requires academic expertise and is costly, while automatic evaluation depends on often low-quality author-written captions. This paper investigates using large language models (LLMs) as a cost-effective, reference-free method for evaluating figure captions. We first constructed SCICAP-EVAL,[1] a human evaluation dataset that contains human judgments for 3,600 scientific figure captions, both original and machine-made, for 600 arXiv figures. We then prompted LLMs like GPT-4 and GPT-3 to score (1-6) each caption based on its potential to aid reader understanding, given relevant context such as figure-mentioning paragraphs. Results show that GPT-4, used as a zero-shot evaluator, outperformed all other models and even surpassed assessments made by Computer Science and Informatics undergraduates, achieving a Kendall correlation score of 0.401 with Ph.D. students' rankings.

## 1 Introduction

There has been increasing interest in automating caption generation for scientific figures in scholarly articles. The SCICAP dataset (Hsu et al., 2021), an extensive collection of scientific figures and captions from arXiv papers, encouraged more work in this area and led to several innovative approaches (Yang et al., 2023; Aubakirova et al., 2023; Tang et al., 2023; Li and Tajbakhsh, 2023). However, despite advances in caption generation for scientific figures, evaluating the results is still challenging. Human evaluation is costly as only domain experts can assess captions, making crowdsourced evaluation impractical. Meanwhile, automatic evaluation is not reliable. Huang et al. (2023) found that

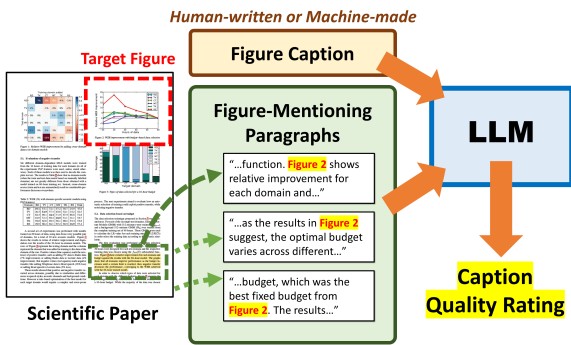

Figure 1: A cost-effective, reference-free evaluation of scientific figure captions using LLMs. We prompted LLMs to score (1-6) a caption based on its potential to aid reader understanding, given relevant context such as figure-mentioning paragraphs.

domain-specific Ph.D. students did not prefer captions rated highly by automatic scores. Fortunately, recent findings in this area may provide solutions to the challenge. Figure captions were found to be similar to summaries of figure-mentioning paragraphs (Huang et al., 2023), *i.e.*, most words in captions can be semantically traced back to these paragraphs, and captions can be effectively generated using text-summarization models. A missing piece from previous research is the use of this finding for evaluation, which we address in this paper.

This paper explores the **cost-effective, reference-free evaluation of scientific figure captions using pre-trained large language models (LLMs)**. Figure 1 overviews the process. The intuition is that since a scientific figure's caption in scholarly articles often functions as a condensed summary of all figure-mentioning paragraphs (Huang et al., 2023), LLMs, when given the appropriate context, can effectively evaluate how well the caption captures the essential information from these contexts (Chiang and Lee, 2023; Shen et al., 2023; Wu et al., 2023).

We first introduced SCICAP-EVAL, a human evaluation dataset for scientific figure captions that contains 600 figures from arXiv papers. For ev-

---

[1]SCICAP-EVAL is accessible at `https://github.com/Crowd-AI-Lab/SciCap-Eval`.

ery figure, we collected six captions: one authored by the original paper authors and five machine-generated. Two participant groups independently assessed these captions. The first group, Ph.D. students from the figure's domain, ranked the six captions for each figure. The second group, undergraduate Computer Science and Informatics students, rated each caption's helpfulness. We used Ph.D. students for their in-depth understanding but recognized their limited availability. Undergraduates were easier to recruit, but their knowledge might be insufficient. We aimed to examine if undergraduates could replace Ph.D. students in building a human evaluation dataset for scientific figure captions. Equipped with SCICAP-EVAL, we then prompted LLMs like GPT-4 and GPT-3.5 to assign a score (1-6) to each caption, based on how helpful they could be for readers to understand the paper. The results showed GPT-4's effectiveness as a zero-shot evaluator. It achieved a 0.401 Kendall correlation with Ph.D. students' rankings, surpassing all other models and even exceeding evaluations by Computer Science and Informatics undergraduates.

This paper presents three contributions. Firstly, we have validated the effectiveness of using LLM to evaluate figure captions, exploring an avenue previously untouched with such techniques. Secondly, we have developed a new dataset, cultivated over a few months with an investment of over $3,300, poised to serve as a benchmark for subsequent research. We recruited twenty undergraduates, each handling 20 batches, with a completion time of 30 minutes to an hour per batch. Additionally, fifteen Ph.D. students participated, each completing four batches, each batch spanning 30 minutes to an hour. Lastly, our work resonates with the broader NLP community's drive to discover fresh applications for LLMs. In cases where human evaluation is costly and reference-based automatic evaluation is unreliable, LLMs offer an efficient alternative.

## 2 Related Work

Assessing scientific figure captions is both costly and time-consuming due to the need for specialized knowledge and understanding of the entire scholarly article. This has previously limited evaluations to a small scale, typically performed by graduate students in the field. For instance, Huang et al. (2023) only managed to employ 3 Ph.D. students with NLP backgrounds to evaluate 90 figures' captions. In a similar vein, Yang et al.

(2023) could only utilize 2 human annotators to evaluate 100 figure captions. Automatic evaluation, on the other hand, has been more broadly used for its cost-effectiveness and ease. Metrics such as BLEU, ROUGH, BLEURT, CIDEr, BERTScore,and MoverScore are common (Kantharaj et al., 2022; Huang et al., 2023; Yang et al., 2023; Chen et al., 2019, 2020; Sellam et al., 2020; Zhang* et al., 2020), using human-written captions as references. However, these metrics are limited, including favoring longer captions (Sun et al., 2019) and relying on often poorly-written human-generated captions. For example, Huang et al. (2023) found that models preferred by humans often scored lower in automatic evaluations. These challenges in evaluating scientific figure captions motivate our pursuit of cost-effective, reference-free evaluation techniques.

## 3 SCICAP-EVAL Dataset

SCICAP (Hsu et al., 2021) is made of over 416,000 line charts (which were referred to as "graph plots" in (Hsu et al., 2021) and 133,543 of them are single-panel) extracted from arXiv papers in Computer Science (cs.*) and Machine Learning (stat.ML). In our paper, we focused on SCICAP's single-panel figures in the NLP, CV, and HCI domains. Most of these figures are 2-dimensional and annotated with axes labels, legends, and other texts. By Arunkumar et al. (2023)'s definition, these figures are "information" visualization rather than "image".

We randomly sampled 600 figures from SCICAP, with each cs.CV, cs.CL, cs.HC domain providing 200 figures. We then obtained six distinct captions for each figure, primarily sourced from Huang et al. (2023) (approach *ii* to *v*), as follows: (*i*) **Author-written captions**; (*ii*) **Pegasus**$_P$, the captions created by the fine-tuned Pegasus model (Zhang et al., 2020), using figure-mentioning paragraphs as the input; (*iii*) **Pegasus**$_{P+O}$, same as Pegasus$_P$ but additionally incorporating figures' OCR as input; (*iv*) **Pegasus**$_O$, same as Pegasus$_P$ but solely relied on OCR as input; (*v*) **TrOCR** (Li et al., 2023), the fine-tuned vision-to-language model; (*vi*) **Template-based approach**, which filled predefined templates with identified axis names.

We then recruited the following two groups of participants to independently annotate the quality of these 3,600 captions. Thus, each caption was assessed twice. The entire study took a few months and cost over $3,300. We engaged 20 undergrad-

uate participants, each responsible for 20 batches. Each batch consisted of 30 figures, and the time taken to finish each batch ranged from 30 minutes to an hour. Additionally, 15 Ph.D. students were recruited, with each student tasked to rank 4 batches. Each batch comprised 10 figures, each having 6 different captions. The time taken by Ph.D. students for each batch varied from 30 minutes to an hour. The authors' institute's Institutional Review Board (IRB) approved this study.

**Human assessments by Ph.D. students on a six-item ranking task.** We recruited 15 Ph.D. students (who are not coauthors of this paper) specializing in CV, NLP, and HCI to rank six captions for each of the 40 figures in their respective fields. Specifically, we formed a ranking task with six items, where each Ph.D. student manually ranked 6 different captions for a figure. These rankings were based on how helpful the captions were in understanding the figures. The annotation interface is shown in Figure 2 in Appendix A. Participants were compensated at $20 per hour, with annotation times varying per individual. To measure data quality, we recruited two non-coauthor Ph.D. students to annotate a specific batch for assessing inter-annotator agreement, which yielded a Kendall of 0.427.

**Human assessments by undergraduate students on a three-class labeling task.** We recruited 20 STEM-focused undergraduate students, primarily Computer Science and Informatics majors,[2] to rate the helpfulness of each caption with a "Yes," "No," or "Unsure" rating. We turned to the three-class labeling task as we realized that the six-item ranking task was too challenging for undergraduate students, which resulted in a long completion time. We understand using different annotation tasks would raise concerns of comparability between Ph.D. and undergraduate students' assessments, but this is the acceptable trade-off to meaningfully collect a broader set of undergraduate students' feedback. For each participant, we assigned five batches covering three different domains, with each batch comprising 30 unique figure captions. In addition to the helpfulness rating, students were asked to identify errors in the extracted figure or caption and label caption features. Full instructions, the user interface, and statistics of the annotation can be found in Appendix B. Compensation was $15

---

[2]Computer Science: 5, Biochemistry: 1, Biology: 1, Statistics: 1, Information Sciences and Technology: 12.

per batch, with annotation times ranging from 30 minutes to one hour. To ensure annotation quality, we provided each participant with a tutorial using an additional 30 figures.

## 4 Methods

In this section, we describe different approaches used for caption rating, including zero-shot, few-shot, and Chain-of-Thought prompting with LLMs, and fine-tuning a classifier using the data collected.

**Zero-Shot and Few-Shot Prompting with LLMs.** We included both the target caption and the figure-mentioning paragraphs from the paper (*e.g.*, "As shown in Figure 4,..."), and then asked the LLM to evaluate the caption's quality. For the few-shot setting, we included three randomly selected captions with the highest and lowest rankings from Ph.D. students' evaluations in the prompt. See Appendix D for the actual prompts.

**Chain-of-Thought Prompting** (Wei et al., 2022) **(QA, QA-YN).** Our intuition is that an effective figure caption distills the essential information from pertinent paragraphs. Acting on this intuition, we implemented a Chain-of-Thought approach to prompt LLMs to calculate evaluation scores. We first provided the LLMs with figure-mentioning paragraphs, asking them to generate questions that a suitable caption should answer. Following this, the LLMs were presented with a caption and asked to identify whether it could answer each question (Yes/No). The final evaluation score of the caption was then derived from the percentage of "Yes" responses. In this paper, we explored two question types: open-ended (QA) and yes/no questions (QA-YN). See Appendix D for the prompts used.

**Classifiers learned from SCICAP-EVAL data.** We fine-tuned the SciBERT classifier (Beltagy et al., 2019) with SCICAP-EVAL data to predict figure captions' helpfulness (Yes/No).

## 5 Experimental Results

**Experimental setups.** We conducted experiments with four LLMs: GPT-4 (OpenAI, 2023), GPT-3.5 (Brown et al., 2020), Falcon (Almazrouei et al., 2023), and LLaMA-2 (Touvron et al., 2023), across various settings. We parsed the prediction scores from the outputs. When the LLMs failed to predict, we allocated the lowest score of 1. For the SciBERT classifier, we split the dataset into

| Model | Setting/ Training Data | (a) Correlation with PhD Students' Reversed Rank [Good to Bad Caption: 6, 5, ..., 2, 1] | | | Correlation with PhD Students' Reciprocal Rank | |
|---|---|---|---|---|---|---|
| | | | | | (b) Original Order (Picking good samples) [Good to Bad Caption: $\frac{1}{1}, \frac{1}{2}, ..., \frac{1}{5}, \frac{1}{6}$] | (c) Reversed Order (Spotting bad samples) [Good to Bad Caption: $\frac{1}{6}, \frac{1}{5}, ..., \frac{1}{2}, \frac{1}{1}$] |
| | | $\rho$ | $\tau$ | $r_s$ | $\rho$ | $\rho$ |
| GPT-4 | Zero-Shot | **.501** | **.401** | **.491** | **.391** | **-.492** |
| | Few-Shot | **.531** | **.429** | **.523** | **.425** | **-.513** |
| | CoT (QA) | .283 | .270 | .315 | .277 | -.223 |
| | CoT (QA-YN) | .357 | .334 | .400 | .324 | -.307 |
| GPT-3.5 | Zero-Shot | .465 | .370 | .462 | .383 | -.433 |
| | Few-Shot | .462 | .371 | .462 | .403 | -.402 |
| | CoT (QA) | .270 | .276 | .347 | .249 | -.226 |
| | CoT (QA-YN) | .365 | .329 | .404 | .317 | -.327 |
| LLaMA 2-70B | Zero-Shot | .407 | .342 | .413 | .326 | -.392 |
| | Few-Shot | .424 | .353 | .430 | .335 | -.405 |
| Falcon-7B | Zero-Shot | .026 | .048 | .055 | .018 | -.030 |
| | Few-Shot | .044 | .048 | .058 | .044 | -.035 |
| Falcon-40B | Zero-Shot | .169 | .137 | .167 | .152 | -.136 |
| | Few-Shot | .150 | .119 | .143 | .126 | -.137 |
| SciBERT* | Undergrad | .329 | .290 | .329 | .261 | -.319 |
| | PhD | .372 | .308 | .379 | .372 | -.372 |
| Human | Undergrad | .221 | .195 | .221 | .206 | -.196 |

Table 1: Correlation coefficients (Pearson $\rho$, Kendall $\tau$, and Spearman $r_s$) of model output ratings versus Ph.D. students' assessments (N=3,159, excluding error cases), based on different rank conversions: (1) Reversed Rank, (2) Reciprocal Rank, and (3) Reversed Reciprocal Rank.*: SciBERT's performance was evaluated on the test split, comprising only 10% of the entire dataset.

80/10/10 for train/validation/test and used the best model from the validation set for final testing. We excluded data marked as errors in undergrads' annotations, resulting in 3,159 valid samples. We fine-tuned SciBERT on undergraduate and Ph.D. annotations, respectively. Captions ranked bottom three by Ph.D. students were labeled as "No" and the top three as "Yes". We treated "No" and "Unsure" as "No" for undergraduate labels. The model training details and decoding configuration are described in Appendix C.

**GPT-4 prompting achieved the highest correlations with Ph.D. students' assessments.** Table 1(a) shows that GPT-4, prompted in either a zero-shot or few-shot manner, exhibited the strongest correlations with Ph.D. students' judgments, surpassing all other models and even undergraduate students' ratings. Meanwhile, despite Yes-No questions yielding better results than open-ended questions, our Chain-of-Thought approaches generally underperformed. We hypothesize that answering questions may only partly determine the helpfulness of a caption. More research is needed to develop improved workflows.

**GPT-4 is better at spotting bad examples than selecting good examples.** We conducted additional analysis to determine whether LLMs as automatic caption evaluators are more effective at identifying good or bad examples. Table 1 displays the Pearson ($\rho$) correlation coefficients between each model's ratings and the reciprocal rank of Ph.D. students. The original order of reciprocal rank (b) places more weight on top selections, with higher scores indicating a model's effectiveness at identifying top examples. Conversely, the reversed order of reciprocal rank (c) prioritizes bottom selections, with higher absolute scores signifying the model's proficiency at pinpointing poor examples. Table 1 shows that GPT-4, in either zero-shot or few-shot prompting, excels at identifying low-quality over high-quality examples. This suggests its utility in cleaning training data by detecting poor samples.

## 6 Discussion

**Do we need paragraphs that mention figures for this approach?** To delve deeper into GPT-4's capability to evaluate captions, we conducted an ablation study to examine the influence of figure-mentioning paragraphs in the prompt. The results,

| | Reversed Rank | | | Reciprocal Rank | | T-Test |
| | (a) | | | (b) | (c) | |
| | $\rho$ | $\tau$ | $r_s$ | $\rho$ | $\rho$ | |
|---|---|---|---|---|---|---|
| **All** | .487 | .399 | .493 | .396 | -.460 | <0.001 |
| **First** | .478 | .392 | .484 | .387 | -.455 | <0.001 |
| **Random** | .479 | .394 | .486 | .395 | -.452 | <0.001 |
| **Caption** | .158 | .125 | .150 | .193 | -.078 | - |

Table 2: Ablation study illustrating the correlations between various inputs and Ph.D. students' rankings. **All**, **First**, and **Random** represent all paragraphs, the first paragraph, and a randomly selected paragraph, respectively. (a), (b), (c) are defined in Table 1. The T-Test assesses the difference in scores when using different inputs versus using only the caption. The results indicate providing paragraphs is necessary.

| | OCR | Visual | Stats | Relation | Takeaway |
|---|---|---|---|---|---|
| **Ph.D.** | **.299** | .120 | .110 | .089 | .195 |
| **Undergrad** | .323 | .279 | .227 | .378 | **.479** |

Table 3: Pearson correlation coefficients ($\rho$) between caption features and the helpfulness ratings.

presented in Table 2, indicate that GPT-4's performance drops a lot when paragraphs are not presented, highlighting the need for paragraphs.

**Toward personalized caption generation.** A goal of having undergraduate students to rate captions was to assess their potential to replace Ph.D. students in constructing human evaluation datasets for scientific captions, given their easier recruitment. However, their "helpfulness" ratings did not correlate well with those of Ph.D. students, suggesting different reader groups may perceive "helpfulness" differently. We further probed this by correlating caption features (annotated by undergraduates, see Appendix B) with the helpfulness assessments by Ph.D. and undergraduate students. Table 3 shows that for Ph.D. students, a caption's helpfulness hinges most on its reference to terms and text from the figure (OCR), while undergraduate students prioritize captions that state the figure's takeaway messages (Takeaway). Furthermore, a brief post-study survey revealed variations among Ph.D. students' assessment criteria. While the accuracy of information was a priority for most, some students focused on the ability of the caption to summarize the figure image, and others evaluated the volume of information conveyed within the caption. These observations suggest that future caption generation systems could focus on personalization to meet the needs of various reader groups.

**Fine-tuning LLMs to predict Helpfulness.** We do believe that fine-tuning LLMs presents a promising direction. However, our preliminary experiments with fine-tuning LLaMA-2 with LoRA (Hu et al., 2022) (both the 7B and 13B models) have not yielded encouraging results thus far. Pearson correlation with Ph.D. Students' Reversed Rank is 0.164 (7B) and 0.183 (13B) respectively. See Appendix C for training details. We believe it requires more exploration in the future.

**How did the appearance and data domain of the images impact perceived utility?** This work focuses on the utility of varied caption text of figures rather than the utility of the figures' images. We have chosen to focus on captions as we believe captions provide the most feasible entry point for technologies to intervene and boost the utility of figures in scholarly articles. Captions, often standalone like the caption{} label in LaTeX, can be revised or replaced without altering the figure image. No raw chart data is strictly needed for improving captions. Using technology to customize or improve caption text is more achievable than altering figure images automatically, especially given the capabilities of LLMs. In the long run, we agree that the utility of the captions can be considered jointly with the figure images– as a good caption would not save a poorly designed visualization– but this is a future direction and beyond this paper's scope.

## 7 Conclusion and Future Work

In conclusion, we have shown the capacity of LLMs, specifically GPT-4, to evaluate scientific figure captions effectively. This work produced SCICAP-EVAL, a human evaluation dataset of 3,600 captions, both original and machine-generated, for 600 arXiv figures. Using SCICAP-EVAL, we confirmed GPT-4's ability to assess caption quality when supplied with context from figure-mentioning paragraphs. GPT-4 even outperformed Computer Science and Informatics undergraduates, achieving a Kendall correlation score of 0.401 against Ph.D. students' ratings. Future work will leverage this evaluation capability to cleanse noisy training data, a well-known challenge in figure caption generation (Huang et al., 2023). Furthermore, we plan to investigate the personalized captions generation to cater to individual reader requirements. Last, we will explore ways to consider caption factuality in the evaluation metrics.

## Acknowledgements

We are grateful to the anonymous reviewers for their insightful feedback and to all the participants in our user study. We would also like to thank Tong Yu from Adobe for his feedback on the draft. This research was made possible through support from Adobe's gift funds and seed funding from Pennsylvania State University's College of Information Sciences and Technology (IST).

## Limitations

Despite its potential, our work is subject to several limitations. Firstly, similar to the method of Huang et al. (2023), our approach requires figure-mentioning paragraphs to act as the context for effective LLM-based caption evaluation. However, mention identification is not always straightforward in real-world data; for example, no mentions were identified for 18.81% of figures in the original dataset. Secondly, our best results were achieved with a closed-source LLM, implying we inherit limitations such as restricted understanding of the model's training and data sources. Thirdly, our evaluation approach does not prioritize verifying the accuracy of the information within a caption, potentially leading to high ratings for inaccurate captions. Finally, as our methodology is solely text-dependent, it cannot capture any figure's visual elements not mentioned in the text.

## Ethics Statement

The proposed technology is considered low-risk as inaccurate figure caption assessments are unlikely to harm readers significantly. However, it is worth noting that our text-based approach inherently overlooks visual content, which could potentially influence the accessibility of the technology in evaluating figure captions.

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

## A  Annotating Interface for Ph.D. Students

Figure 2 shows the ranking task interface used by the Ph.D. students in Section 3.

## B  Annotating Instruction for Undergraduate Students

Figure 3 shows the rating task interface used by the undergraduate students in Section 3. We inquired whether the figure-caption combination contained any of the following four errors (Yes/No):

- Image Extraction Error: The image we extracted from the PDF (shown above) has obvious errors (e.g., not containing the complete figure, containing parts that are not figures, damaged image, etc.)

- Text Extraction Error: The text we extracted from the PDF (shown above) has obvious errors (e.g., not containing the complete caption, containing extra text that is not the caption, incorrect text recognition, etc.)

- Not a Line Chart: This figure is not a line chart.

- Compound Figure: This figure is a compound figure that contains multiple subfigures.

The resulting error counts are shown in Table 4. We also asked participants to annotate whether the caption contains any of the following six aspects (Yes/No):

- OCR (Optical Character Recognition): Does this caption mention any words or phrases that appear in the figure? (Examples include the figure title, X or Y axis titles, legends, names of models, methods, subjects, etc.)

- Visual: Does this caption mention any visual characteristics of the figure? (Examples include color, shape, direction, size, position, or opacity of any elements in the figure.)

- Stats: Does this caption mention any statistics or numbers from the figure? (For example, "20% of..." or "The value of .. is 0.33...".)

- Relation: Does this caption describe a relationship among two or more elements or subjects in the figure? (For example, "A is lower than B," "A is higher than B," or "A is the lowest/highest in the figure.")

| Error Type | Total |
|---|---|
| Image Extraction Error | 102 |
| Text Extraction Error | 242 |
| Not a Line Chart | 101 |
| Compound Figure | 23 |

Table 4: For all 3,600 figure-caption pairs, there are 3,159 valid samples. The remaining 441 figure-captions contain at least one error.

- Takeaway: Does this caption describe the high-level takeaways, conclusions, or insights the figure tries to convey?

- Helpfulness: Does this caption help you understand the figure?

## C  Training and Decoding Details

We describe the model training details and the decoding configuration used in Section 5.

**Training Details for Classification models.** We fine-tune SciBERT[3] checkpoint from HuggingFace for 100 epochs, using batch size = 32, learning rate = 5e-5 with a linear decay scheduler, warmup steps = 500, weight decay = 0.01. We evaluate every 100 steps, and the checkpoint with the highest f1 on validation set is kept and used to predict final result.

For our fine-tuning LLaMA-2, we modeled it as a text generation task. The input includes information from figure-mentioning paragraphs and target captions, and the model outputs evaluation for each aspect in a specific format (*e.g.*, [helpfulness: yes]). The model was trained with a lora_rank = 8, learning rate = 5e-5, batch size = 16, and we trained the model for 50 epochs. We kept the last checkpoint for evaluation.

**Decoding Details for Open Large Language models.** Parameter settings for Falcon-7b,[4] Falcon-40b,[5] and LLaMA 2-70b[6] are the same: do_sample = True, top_p = 0.95, min_new_tokens = 10, max_new_tokens = 200, repetition_penalty = 1.0, temperature = 0.1, num_beams = 1. Note that all the models used in the experiment are the instruction-following model.

---

[3]We used `allenai/scibert_scivocab_uncased`.
[4]We used `tiiuae/falcon-7b-instruct`.
[5]We used `tiiuae/falcon-40b-instruct`.
[6]We used `meta-llama/Llama-2-70b-chat-hf`.

# D Prompts used for LLMs

In this section, we provide the prompt we used in Section [4]. `[PARAGRAPHS]` and `[CAPTION]` are placeholders for figure-mentioning paragraphs and the target caption.

**Zero-shot prompting.** *"Given the paragraphs and caption below, please rate the level of usefulness of the caption from 1 to 6 based on how well the caption could help readers understand the important information. 6 is the highest; 1 is the lowest. Please also explain your rating. Paragraphs: [PARAGRAPHS]. Caption: [CAPTION]"*

**Few-shot prompting.** *"Given the paragraph and caption below, please rate the level of usefulness of the caption from 1 to 6 based on how well the caption could help readers understand the important information. 6 is the highest; 1 is the lowest. Please also explain your rating. The following are 3 examples of high-quality captions: Best-1, Best-2, Best-3. The following are 3 examples of low-quality captions: Worst-1, Worst-2, Worst-3. Paragraphs: [PARAGRAPHS]. Caption: [CAPTION]"*

**Chain-of-Thought prompting.** Here are the prompts used for generating questions and obtaining answers for each question. We explore two types of questions, open-ended (QA) and yes/no questions (QA-YN), and prompts only difference including **yes or no** in question generation:

- **Open-ended Question Generation:** *"The following are paragraphs from a paper that mentioned figure-index. Based on these paragraphs, please generate at most five questions that the caption of figure-index should be able to answer. These questions quite be interesting and useful to the readers of the paper, who are mostly researchers in domain and AI."*

- **Yes/No Question Generation:** *"The following are paragraphs from a paper that mentioned figure-index. Based on these paragraphs, please generate at most five yes or no questions that the caption of figure-index should be able to answer. These questions quite be interesting and useful to the readers of the paper, who are mostly researchers in domain and AI."*

- **Answer:** *"The following is the caption of figure-index. Does this caption answer each question? Please answer Yes or No one by one and explain why or why not. Do not repeat the question."*

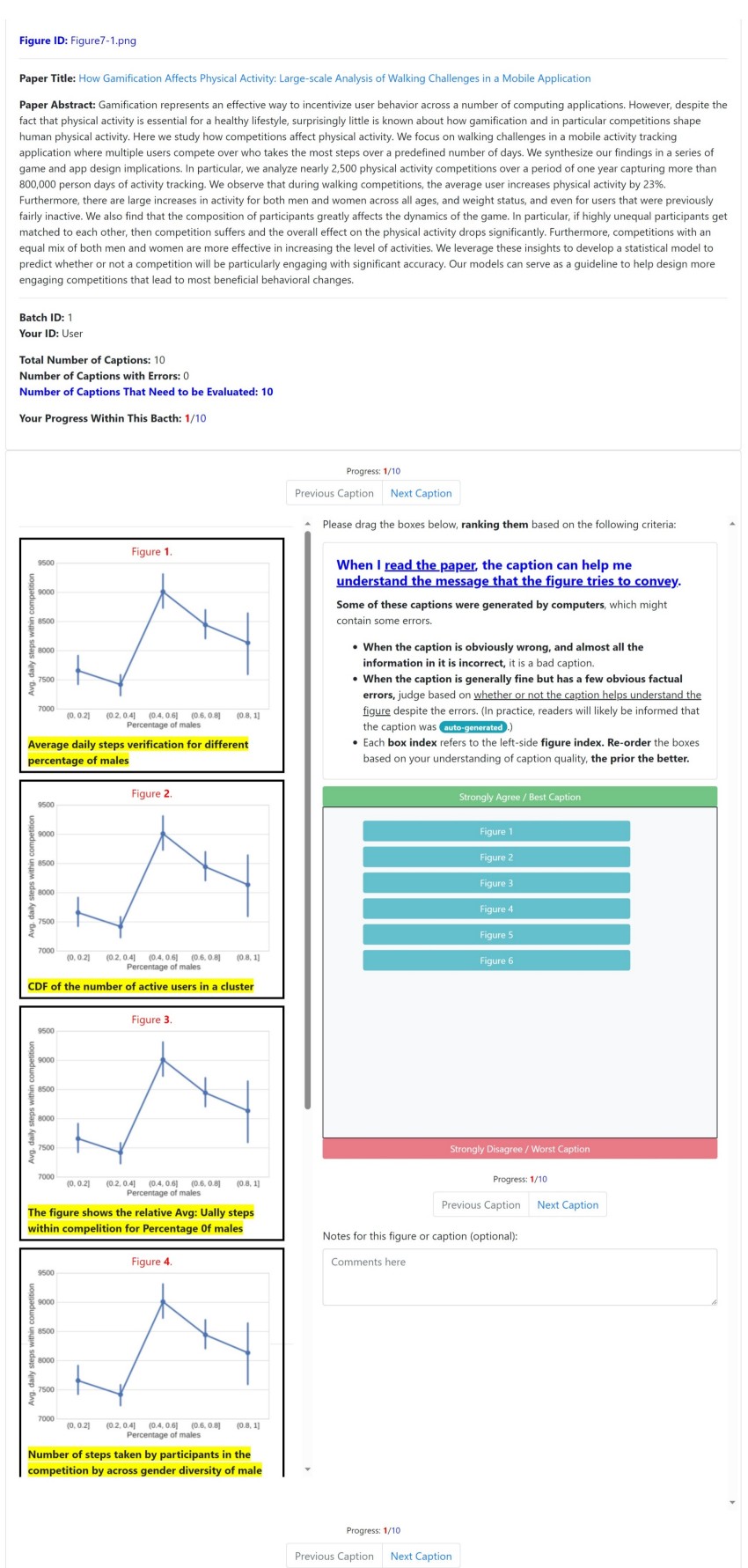

Figure 2: The interface used by the Ph.D. students to rank the captions.

# Figure Captioning Annotation

- In this HIT, you will see **10** figure-caption pairs about scientific articles. Read each figure and select the best answer for each question. Please spend at least 30 seconds on each one.

**Please look at this example figure. Below are example captions with sample questions you will be asked about the pairing of the figure and various captions.** [hide examples]

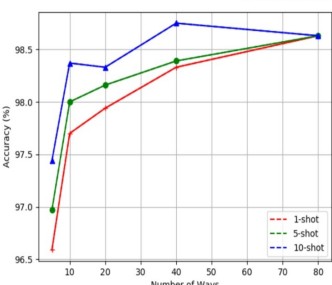

1. **Does this caption mention any words or phrases that appear in the figure?** (Examples include the figure title, X or Y axis titles, legends, names of models, methods, subjects, etc.)
   **Example caption:** Figure 6: ==Accuracy== of our proposed model in different ==N-way K-shot== tasks.
   **Answer:** Yes: The caption contains words in the figure (highlighted in yellow).
   **Example caption:** Figure 6: There are three lines in the graphplot.
   **Answer:** No: The caption doesn't mention any words that appear in the figure.

2. **Does this caption mention any visual characteristics of the figure?** (Examples include color, shape, direction, size, position, or opacity of any elements in the figure.)
   **Example caption:** Figure 6: The 10-shot **(blue line)** with ==jagged shape== is ==at the top== of the figure.
   **Answer:** Yes: The caption mentions color, shape, and position in the figure.
   **Example caption:** Figure 6: The relation between accuracy and number of ways.
   **Answer:** No: The caption doesn't mention any visual characteristics.

3. **Does this caption mention any statistics or numbers from the figure?** (For example, "20% of..." or "The value of .. is 0.33...".)
   **Example caption:** Figure 6: 40-way 10-shot achieves almost ==99%== of accuracy.
   **Answer:** Yes: The caption mentions statistics in the figure.
   **Example caption:** Figure 6: The blue line (10-shot) has the highest accuracy.
   **Answer:** No: There are no statistics in the caption from the figure.

4. **Does this caption describe a relationship among two or more elements or subjects in the figure?** (For example, "A is lower than B," "A is higher than B," or "A is the lowest/highest in the figure.")
   **Example caption:** Figure 6: The 10-shot overall has the ==highest== accuracy compared to 1-shot and 5-shot.
   **Answer:** Yes: The caption describes a relationship among elements in the figure.
   **Example caption:** Figure 6: There are three lines in the graphplot.
   **Answer:** No: There are no descriptions of relationships among elements in the caption.

---

| Annotator Name | Please Enter Your Name Here. |
| --- | --- |

---

Progress: **1**/10

[Previous Caption] [Next Caption]

**Please read the following caption and figure image carefully and answer the questions on the side.**

==Plots of the query time vs construction time tradeoff for Hierarchical NSW on 10M SIFT dataset.==

**Does the image or caption shown above have any issues?**

- ☐ **Image Extraction Error:** The image we extracted from the PDF (shown above) has obvious errors (e.g., not containing the complete figure, containing parts that are not figures, damaged image, etc.)
- ☐ **Text Extraction Error:** The text we extracted from the PDF (shown above) has obvious errors (e.g., not containing the complete caption, containing extra text that is not the caption, incorrect text recognition, etc.)
- ☐ **Not a Line Chart:** This figure is not a line chart.
- ☐ **Compound Figure:** This figure is a compound figure that contains multiple subfigures.

**1. Does this caption mention any words or phrases that appear in the figure?** (Examples include the figure title, X or Y axis titles, legends, names of models, methods, subjects, etc.)
- ○ Yes
- ○ No
- ○ Not sure

**2. Does this caption mention any visual characteristics of the figure?** (Examples include color, shape, direction, size, position, or opacity of any elements in the figure.)
- ○ Yes
- ○ No
- ○ Not sure

Figure 3: The interface used by the undergraduate students to rate the captions.