# OpenReview forum: "GPT-4 as an Effective Zero-Shot Evaluator for Scientific Figure Captions"
_EMNLP/2023/Conference — EMNLP 2023 Findings_

### Official Review · Reviewer_whcN · 2023-07-31

**Soundness:** 3

**Excitement:**

3: Ambivalent: It has merits (e.g., it reports state-of-the-art results, the idea is nice), but there are key weaknesses (e.g., it describes incremental work), and it can significantly benefit from another round of revision. However, I won't object to accepting it if my co-reviewers champion it.

**Missing References:**

Bertscore: Evaluating text generation with bert. Zhang et al. 2019.
BLEURT: Learning Robust Metrics for Text Generation. Sellam et al. 2020.

**Paper Topic And Main Contributions:**

The paper investigates using LLM to evaluate the quality of generated image caption. Specifically, the authors first collected a image caption dataset incorporating 3,600 <image, caption> pair from both human written and machine produced ones. Then they applied LLM to score the quality of caption, which is compared with the one from Ph.D. students and undergraduates, and found that GPT4 performs well as it achieves highest correlation score with Ph.D. students' assessments in all settings.

**Questions For The Authors:**

I also have the following questions for the authors:
1. You hire Ph.D students from the expert domain according to the specific image, while undergraduate students only from CS department. I understand that undergraduate students may not have the specific research area. But I still think this setting might be a little unfair to achieve the final conclusion. In order the get the general conclusion, is it better to hire undergraduate students from diverse departments to achieve more general conclusion?
2. There seems no consistent conclusion. The author claimed they explore cost-effective evaluation, but GPT4 is expensive to use. While for their own tuned, cost-free SciBERT, it seems the performance lays far behind GPT4. Is there any final conclusion for this paper to give that which evaluator achieves high performance and is really cost-effective?

**Reasons To Accept:**

The paper proved that GPT4 serves as an excellent image caption evaluator even in zero-shot setting.

**Reasons To Reject:**

There are no too much novel techniques in this paper, and the whole pipeline seems like a experimental report without any technical contribution.

**Reproducibility:**

4: Could mostly reproduce the results, but there may be some variation because of sample variance or minor variations in their interpretation of the protocol or method.

**Reviewer Confidence:**

4: Quite sure. I tried to check the important points carefully. It's unlikely, though conceivable, that I missed something that should affect my ratings.

---

> ### Author Rebuttal · Authors · 2023-08-29
>
> We thank Reviewer whcN for the valuable feedback and for recognizing our approach as an excellent caption evaluator.
>
> #### **[Contribution]**
> We believe the proposed approach has value to the NLP community because it is easy to use and gives strong results. At a higher level, our work responds to the NLP community’s pressing research theme of exploring new uses of LLMs. Furthermore, our work responds to a set of newly-emerged findings: Figure captions were recently found to be similar to summaries of figure-mentioning paragraphs [1], i.e., most words in captions can be semantically traced back to these paragraphs (Section 4 of [1]), captions can be effectively generated using text-summarization models (Section 6 of [1].) One missing piece from prior work is to exploit this finding for evaluation, which we proposed in this paper. We will revise the paper to make this contribution more explicit.
>
> In addition to the technical contribution, we also made artifact contributions by creating a new dataset, which will be released publicly upon publication. The recruiting and annotation process alone took about 4-5 months and cost over $3,300. Other researchers will be able to build new work on top of this dataset.
>
> #### **[Questions (Q)]**
> **[Q1 - Background of the Undergraduate Students]**:
>
> To clarify, the undergraduate students we recruited were from several STEM majors, just that many of them were in CS or Informatics. Please see the table below for the detailed distribution.
>
> | Major                                  | Number of Undergrad Students |
> |----------------------------------------|:------------------------------:|
> | Computer Science                       | 5                            |
> | Biochemistry                           | 1                            |
> | Biology                                | 1                            |
> | Data Science                           | 3                            |
> | Security and Risk Analysis             | 1                            |
> | Cybersecurity Analytics and Operations | 5                            |
> | Statistics                             | 1                            |
> | Enterprise Technology Integration      | 1                            |
> | Information Sciences and Technology    | 2                            |
>
>
> **[Q2 - Cost of GPT-4]**:
>
> By “cost-effective,” we meant that our method, though not free like some automatic evaluation methods, remains more economical than human evaluations. Evaluating scientific figure captions requires participants to comprehend scientific papers, making it hard to recruit competent participants (e.g., can not use MTurk) and generally expensive to run human evaluations. In our work, we paid each undergraduate student \\$15 to label 30 figures (30 figure-caption pairs); we paid each Ph.D. student \\$20 per hour, and it took them 1.5 to 4 hours to rank 6 captions for each of 40 figures (40*6 = 240 figure-caption pairs). GPT-4, with OpenAI’s current price rate, costs \\$40&ndash;\\$50 to evaluate 3,600 figure-caption pairs in our study. While we agree that human evaluation is important and should not be ignored or replaced by automatic evaluations, GPT-4 offers an easy and affordable option for quick sanity checks, allowing practitioners to iterate their caption system quickly before having humans test it.
>
> #### **[References]**
>
> [1] Huang, C. Y., Hsu, T. Y., Rossi, R., Nenkova, A., Kim, S., Chan, G. Y. Y., ... & Huang, T. H. K. (2023). Summaries as Captions: Generating Figure Captions for Scientific Documents with Automated Text Summarization. INLG 2023.

---

### Official Review · Reviewer_Tt8o · 2023-08-04

**Soundness:** 3

**Excitement:**

3: Ambivalent: It has merits (e.g., it reports state-of-the-art results, the idea is nice), but there are key weaknesses (e.g., it describes incremental work), and it can significantly benefit from another round of revision. However, I won't object to accepting it if my co-reviewers champion it.

**Missing References:**

Line 53 -- ROUGE score citation

Line 131 -- Pegasus

Line 135 -- TrOCR

Line 195 -- Chain-of-thought

**Paper Topic And Main Contributions:**

The paper proposes SciCap-Eval, which is a labelled dataset which contains scores for captions taken from SciCap (Hsu et al, 2021). SciCap-Eval contains both human and machine-generated scores for 3,600 scientific captions (ranging over 600 figures). The main issue SciCap-Eval is addressing is the difficulty of evaluating scientific captions, and they solve this by providing annotated scores.

Each figure in SciCap-Eval comes with 6 captions (1 human ground truth and 5 machine-generated). These were then assigned the PhD students (who are experts in the field of the figures) to score the captions from 1 to 6. They treat these PhD captions as the new ground truth.

The authors then evaluate various captioning methods such as GPT-4, GPT-3, Falcon, SciBERT (fine-tuned), and undergraduate students. They measure the correlation of these scores with the scores given by the PhD students and conclude that GPT-4 (both zero-shot and few-shot) has the highest correlation, outperforming even fine-tuned models and undergraduate students.

**Questions For The Authors:**

- How did you sample figures from SciCap? What criteria did you use? Was it just random?
- Details for the models: What size of Falcon? Was this instruction-tuned?
- You fine-tuned SciBERT, but why didn't you fine-tune Falcon? There are a lot of ways to fine-tune large models now, especially with tools like 8-bit and LoRA
- Is there a reason you chose to score from 1 to 6? 6 seems like a rather arbitrary number.

Other comments:
- I think some of the questions asked to the undergraduate students (e.g. types of errors) are very interesting! It would be nice if you can also add some statistics to the Appendix so we can get an idea of what the dataset is like.
- Any thoughts on how to evaluate caption factuality? I understand it's challenging but I think it's an interesting and important issue especially for scientific captions, I was just curious if you had any thoughts.

**Reasons To Accept:**

A lot of papers right now are using GPT-4 as a ground truth oracle evaluator (rather than using human annotations), especially for generative tasks. There is a big question mark surrounding this practice because we don't fully know whether this is something GPT-4 is capable of performing well. This paper investigates that topic specifically for scientific captions. As this trend of "oracle GPT-4" continues, I think we need more papers to really show us what GPT-4 can or cannot do, and I think this paper would make a really nice addition to that list.

Even though there are a few gaps in the paper/methodology, I think the main points are solid, and I don't doubt any of the conclusions. I think it makes sense that GPT-4 and GPT-3.5 would perform the best on this task, and it's nice to see it verified.

**Reasons To Reject:**

The biggest elephant in the room is that the paper's "captioning" isn't really captioning -- it's more of summarizing. The models aren't evaluating a caption based on the figure. They're evaluating a caption based on some text, which can potentially miss a lot of information, especially if the text is not very descriptive. I also feel that using a text-only method somewhat takes away from the excitement and difficulty of the task since evaluating a caption based on a scientific figure is a lot more challenging than evaluating a one-sentence summary of a piece of text.

Comparisons in the main table (Table 1) aren't entirely fair. For instance, GPT4 was asked to rate models from 1 to 6, while the undergraduate students were asked to give qualitative discrete values ("Yes"/"No"/"Unsure"), so it is a bit difficult to compare.

Writing is not very thorough. There are a bunch of missing citations, and a lot of the more specific experimental details (e.g. hyperparameters, model sizes) seem to be omitted. Some experimental choices don't seem particularly justified (see Questions below).

**Reproducibility:**

3: Could reproduce the results with some difficulty. The settings of parameters are underspecified or subjectively determined; the training/evaluation data are not widely available.

**Reviewer Confidence:**

4: Quite sure. I tried to check the important points carefully. It's unlikely, though conceivable, that I missed something that should affect my ratings.

**Typos Grammar Style And Presentation Improvements:**

Line 66 -- missing punctuation

Line 127 -- "provides" --> "providing"

---

> ### Author Rebuttal · Authors · 2023-08-29
>
> We thank Reviewer Tt8o for the valuable feedback, which we will incorporate into our final draft. We will add the references and also hire an editor for proofreading.
>
> #### **[Contribution and Limitations of Text-Only Approaches]**
> The rationale behind a text-only approach is that figure captions (in arXiv papers) were recently found to be similar to summaries of figure-mentioning paragraphs [1], i.e., most words in captions can be semantically traced back to these paragraphs (Section 4 of [1]), captions can be effectively generated using text-summarization models (Section 6 of [1].) One missing piece from prior work is to exploit this interesting finding for evaluation, which we proposed in this paper. We will revise the paper to make this contribution more clear.
>
> We are aware of the limitations of text-only approaches for a vision-and-language task, as stated in the Limitation section. We believe our proposed approach has value to the research community regardless of its limitations because it is very easy to use and gives strong results. Exploring the new uses of LLMs is also a pressing topic in the current NLP community. We will incorporate these considerations into the Limitation section.
>
> #### **[Different Labeling Schemes Between PhD and Undergraduate Students]**
> In fact, we initially used the identical 6-item ranking task for both undergraduate and Ph.D. students. However, we soon found that comparing six different captions while comprehending the figure image and the paper content was too challenging for undergraduate students, who performed the task very slowly. After some design iterations, we eventually settled on the current 3-class labeling task for the undergrads.
>
> We understand using different annotation tasks would raise concerns of comparability between Ph.D. and undergraduate students’ assessments, but this is the trade-off we are willing to make in order to meaningfully collect a broader set of undergraduate students' feedback. We will make this decision clear in the final draft.
>
> #### **[Questions (Q)]**
>
>
> #### **[Q1 - Figure Sampling]**
> Yes, we randomly sampled 200 figures from the SciCap dataset’s CV, NLP, and HCI domain (according to the arXiv category,) respectively.
>
> #### **[Q2 - Details of the Models]**
>
> == **Scibert** ==
>
> We finetune “allenai/scibert_scivocab_uncased” checkpoint from HuggingFace for 100 epochs, batch size = 32, learning rate = 5e-5, warmup steps = 500, weight decay = 0.01, we use the best f1 checkpoint on validation set to predict final result.
>
> == **Falcon & LLama2** ==
>
> Given the limited resources, we report Falcon-7b zero-shot and few-shot results before the deadline. But we did try Falcon-40b and Llama2-70b afterward. Parameter settings for both Falcon-7b, Falcon-40b and Llama2-70b are the same. The only difference is the pretrained model. All the model used in the experiment is the instruction-following model.
>
> Falcon-7b: tiiuae/falcon-7b-instruct
>
> Falcon-40b: tiiuae/falcon-40b-instruct
>
> Llama2-70b: meta-llama/Llama-2-70b-chat-hf
>
> Parameter settings: do_sample=True, top_p=0.95, min_new_tokens=10, max_new_tokens=200, repetition_penalty=1.0, temperature=0.1, num_beams=1
>
> == **Correlation Scores** ==
>
> | Model      | Setting | (a) Corr. to PhD Reversed Rank, (6~1)      | (b) Corr. to PhD Original (1~⅙) | (c) Corr. to PhD Reversed (⅙~1) |
> |------------|-----------------------|---------------------------------------|:---------------------------------|:---------------------------------|
> |            |                       | Pear. &nbsp;&nbsp; Kendall. &nbsp;&nbsp; Spear.                              | Pear.                           | Pear.                           |
> | Falcon-40b | zero-shot             | .169 &nbsp;&nbsp;&nbsp;&nbsp;&nbsp; .137 &nbsp;&nbsp;&nbsp;&nbsp;&nbsp;&nbsp;&nbsp;&nbsp; .167    | .152                            | -.136                           |
> |            | few-shot              | .150  &nbsp;&nbsp;&nbsp;&nbsp;&nbsp;  .119   &nbsp;&nbsp;&nbsp;&nbsp;&nbsp;&nbsp;&nbsp;&nbsp;   .143                                | .126                            | -.137                           |
> | Llama2-70b | zero-shot             | .407  &nbsp;&nbsp;&nbsp;&nbsp;&nbsp; .342  &nbsp;&nbsp;&nbsp;&nbsp;&nbsp;&nbsp;&nbsp;&nbsp;   .413                               | .326                            | -.392                           |
> |            | few-shot              | .424      &nbsp;&nbsp;&nbsp;&nbsp;&nbsp;  .353  &nbsp;&nbsp;&nbsp;&nbsp;&nbsp;&nbsp;&nbsp;&nbsp;  .430                         | .335                               | -.405                           |
>
>
> #### **[Q3 - Fine-tuning LLM]**
> We do believe that fine-tuning LLMs presents a promising direction for improving evaluation performance in the future. However, our preliminary experiments with fine-tuning LLaMA-2 (both the 7B and 13B models) have not yielded encouraging results thus far.
>
> For our fine-tuning process, we modeled it as a text generation task. The input includes information from figure-mentioning paragraphs and target captions, and the model outputs evaluation for each aspect in a specific format (e.g., [[helpfulness: yes]]). The model was trained with a lora_rank of 8, a learning rate of 5e-5, batch size of 16, and we trained the model for 50 epochs. We kept the last checkpoint for evaluation. The scores were as follows:
>
> | Model       | Training data | (a) Corr. to PhD Reversed Rank, (6~1)     | (b) Corr. to PhD Original (1~⅙) | (c) Corr. to PhD Reversed (⅙~1) |
> |-------------|-----------------------|---------------------------------------|---------------------------------|---------------------------------|
> |            |                       | Pear. &nbsp;&nbsp; Kendall. &nbsp;&nbsp; Spear.                              | Pear.                           | Pear.                           |
> | *llama2-7b  | Undergrad             | .164 &nbsp;&nbsp;&nbsp;&nbsp;&nbsp; .144 &nbsp;&nbsp;&nbsp;&nbsp;&nbsp;&nbsp;&nbsp;&nbsp; .164 | .162                            | -.138                           |
> | *llama2-13b | Undergrad             | .183 &nbsp;&nbsp;&nbsp;&nbsp;&nbsp; .162 &nbsp;&nbsp;&nbsp;&nbsp;&nbsp;&nbsp;&nbsp;&nbsp;  .183 | .138                            | -.193                           |
>
> *: finetune llama2 classifier performance was evaluated on the test split, comprising only 10% of the entire dataset.
>
> Our computation resources do not allow us to fine-tune LLaMA-2 70B as the training requires several hundreds of hours to run using LoRA and 8-bit quantization. While the initial results for LLaMA-2 7B and 13B were not as we expected, we still believe it is a promising direction to explore. We will add this to our future work.
>
> #### **[Q4 - Range of the Scores]**
> To clarify, it was a ranking task with 6 items, where each Ph.D. student manually ranked 6 different captions for a figure.
>
> #### **[Other Comments (C)]**
> **[C1 - Statistics of the Annotation]**
>
> We will release the student annotation dataset in the final version. For all 3,600 figure-caption pairs, there are 3,159 valid samples. The remaining 441 figure-captions contain at least one error. We give the detailed error analysis below.
>
> |       | ImageError | CaptionError | ClassificationError | SubfigureError |
> |-------|:------------:|:--------------:|:---------------------:|:----------------:|
> | Total | 102        | 242          | 101                 | 23             |
>
> **[C2 - Evaluating Caption Factuality]**
>
> We believe that human evaluation is a promising way to check the caption factuality, but it’s extremely hard to find domain experts to do evaluation.
>
> Another way to check the caption factuality is to align the information between captions and figures/paragraphs. For information whose source information can be successfully identified, its factuality can be guaranteed. Huang et al. [1] used awesome-alignment to extract information shared between captions and paragraphs, which we believe is also a promising way for evaluating factuality. However, the alignments between figures and captions remain unexplored. Although it might be possible to directly align the patches and captions (using the transformer-based visual models), we are not sure whether such small patches can capture concepts such as “the blue line outperforms the red line”. We will add this to our discussion and leave it for future work.
>
> #### **[References]**
>
> [1] Huang, C. Y., Hsu, T. Y., Rossi, R., Nenkova, A., Kim, S., Chan, G. Y. Y., ... & Huang, T. H. K. (2023). Summaries as Captions: Generating Figure Captions for Scientific Documents with Automated Text Summarization. INLG 2023.

---

### Official Review · Reviewer_EmJc · 2023-08-05

**Soundness:** 3

**Excitement:**

3: Ambivalent: It has merits (e.g., it reports state-of-the-art results, the idea is nice), but there are key weaknesses (e.g., it describes incremental work), and it can significantly benefit from another round of revision. However, I won't object to accepting it if my co-reviewers champion it.

**Paper Topic And Main Contributions:**

This paper investigates using large language models for evaluating figure captions. 600 scientific figures were used from SciCap and 6 captions for each figure were generated using different methods. Caption quality was then assessed in a user study with PhD students and undergraduate students. Evaluation of GPT-4, GPT-3.5, and SciBERT was done. GPT-4 prompting achieved the highest correlations with Ph.D. students’ assessments. However, undergraduate “usefulness” ratings did not correlate well with those of Ph.D. students. The authors discuss how this might arise from the difference in perceived utility.

**Questions For The Authors:**

Could you discuss how the appearance and data domain of the images impacted perceived utility more thoroughly? These references might be useful for the same:
What Makes a Visualization Memorable? Borkin et al., 2016 (https://vcg.seas.harvard.edu/publications/what-makes-a-visualization-memorable/paper/)
Image or Information? Examining the Nature and Impact of Visualization Perceptual Classification. Arunkumar et al., 2023. (https://arxiv.org/pdf/2307.10571.pdf)

**Reasons To Accept:**

This paper aims to solve an interesting problem, particularly the use of scientific figures also opens up avenues to fine-tune scientific figure generation.

**Reasons To Reject:**

The approach used is straightforward, however I would have liked more discussion on the attributes of the dataset used, particularly because in data visualization, it has been shown that the presence and absence of design elements, particularly narrative elements in charts heavily impact how the chart utility varies.

**Reproducibility:**

4: Could mostly reproduce the results, but there may be some variation because of sample variance or minor variations in their interpretation of the protocol or method.

**Reviewer Confidence:**

3: Pretty sure, but there's a chance I missed something. Although I have a good feel for this area in general, I did not carefully check the paper's details, e.g., the math, experimental design, or novelty.

**Typos Grammar Style And Presentation Improvements:**

-

---

> ### Author Rebuttal · Authors · 2023-08-29
>
> We thank Reviewer EmJc for pointing out these very interesting papers, which we will cite in our final draft.
>
> #### **[Attributes of Figures Used in the Paper]**
> SciCap [1] is made of over 416,000 line charts (133,543 are single-panel) (which were referred to as “graph plots” in [1]) extracted from arXiv papers in Computer Science (cs.*) and Machine Learning (stat.ML). In our paper, we further focused on SciCap’s single-panel figures in the NLP, CV, and HCI domains. Our impression is that most of these figures are 2-dimensional with axes and some text annotations like legends. These figures should be better characterized as “information” visualization than “image” [2]. We will add more details on our figures' common design elements and attributes in the final draft.
>
> #### **[Utility of Figures vs. Figure Captions]**
> Our paper focuses on the utility of varied caption text of figures rather than the utility of the figures' images. We have chosen to focus on captions as we believe captions provide the most feasible entry point for technologies to intervene and boost the utility of figures in scholarly articles. Captions, often standalone like the \caption{} label in Latex, can be revised or replaced without altering the figure image. No raw chart data is strictly needed for improving captions. Using technology to customize or improve caption text is more achievable than altering figure images automatically, especially given the capabilities of LLMs.
>
> In the long run, we agree that the utility of the captions can be considered jointly with the figure images &ndash;as a good caption would not save a poorly designed visualization&ndash; but we view this as a future direction and beyond this paper's scope.
>
>
> #### **[References]**
> [1] Hsu, T. Y., Giles, C. L., & Huang, T. H. (2021, November). SciCap: Generating Captions for Scientific Figures. In Findings of the Association for Computational Linguistics: EMNLP 2021 (pp. 3258-3264).
>
> [2] Arunkumar, A., Padilla, L., Bae, G. Y., & Bryan, C. (2023). Image or Information? Examining the Nature and Impact of Visualization Perceptual Classification. IEEE Vis 2023.

---

### Meta-Review · Area_Chair_GRtY · 2023-09-17

**Recommendation:** 3

**Metareview:**

The paper presents an investigation into the use of large language models (LLMs) as evaluators for figure captions, specifically scientific ones. Using the SCICAP-EVAL dataset, a human evaluation dataset that contains human judgments for 3,600 scientific figure captions, both original and machine-made, for 600 arXiv figures, the authors find that GPT-4, used as a zero-shot evaluator, outperforms all other models and even surpasses assessments made by computer science undergraduates.

The reviewers have shared diverse views on the paper. Overall, the reviewers acknowledge the paper's merits in investigating an interesting problem, providing a new dataset (SCICAP-EVAL), and demonstrating the effectiveness of GPT-4 as a caption evaluator. The use of GPT-4 as an evaluator, especially in the zero-shot setting, is noteworthy. However, the reviewers also express concerns: One primary concern is the lack of novel techniques in the paper, and the overall approach is seen as more of an experimental report.  There's also concern about the methodology discussed in the paper on figure captioning, which could potentially be misleading as the model doesn't utilize visual input (i.e., the figure) to perform this task. Comparison between the GPT-4 scores and undergraduate students' evaluations is another area of concern due to differing scoring scales and a lack of detail about the experimental design and the rationale behind certain choices. The authors are urged to provide more clarity on these aspects.

In conclusion, while the paper provides valuable insights into the use of LLMs as evaluators for figure captions, there are points raised by the reviewers that need to be addressed, particularly in terms of technical contributions and experimental details. The paper could significantly benefit from another round of revision to address these concerns and questions. However, the reviewers are generally in agreement that the paper has merits and potential contributions to the filed.

---

### Decision · Program_Chairs · 2023-10-07

**Decision:**

Accept-Findings

**Comment:**

The paper presents an investigation into the use of large language models (LLMs) as evaluators for figure captions, specifically scientific ones. Using the SCICAP-EVAL dataset, a human evaluation dataset that contains human judgments for 3,600 scientific figure captions, both original and machine-made, for 600 arXiv figures, the authors find that GPT-4, used as a zero-shot evaluator, outperforms all other models and even surpasses assessments made by computer science undergraduates.

The reviewers have shared diverse views on the paper. Overall, the reviewers acknowledge the paper's merits in investigating an interesting problem, providing a new dataset (SCICAP-EVAL), and demonstrating the effectiveness of GPT-4 as a caption evaluator. The use of GPT-4 as an evaluator, especially in the zero-shot setting, is noteworthy. However, the reviewers also express concerns: One primary concern is the lack of novel techniques in the paper, and the overall approach is seen as more of an experimental report.  There's also concern about the methodology discussed in the paper on figure captioning, which could potentially be misleading as the model doesn't utilize visual input (i.e., the figure) to perform this task. Comparison between the GPT-4 scores and undergraduate students' evaluations is another area of concern due to differing scoring scales and a lack of detail about the experimental design and the rationale behind certain choices. The authors are urged to provide more clarity on these aspects.

In conclusion, while the paper provides valuable insights into the use of LLMs as evaluators for figure captions, there are points raised by the reviewers that need to be addressed, particularly in terms of technical contributions and experimental details. The paper could significantly benefit from another round of revision to address these concerns and questions. However, the reviewers are generally in agreement that the paper has merits and potential contributions to the filed.